# Genome-Wide Characterization of the *BTB* Gene Family in Poplar and Expression Analysis in Response to Hormones and Biotic/Abiotic Stresses

**DOI:** 10.3390/ijms25169048

**Published:** 2024-08-21

**Authors:** Jing Yue, Xinren Dai, Quanzi Li, Mingke Wei

**Affiliations:** 1State Key Laboratory of Tree Genetics and Breeding, Chinese Academy of Forestry, Beijing 100000, China; yuejing@caf.ac.cn (J.Y.); xinrend@caf.ac.cn (X.D.); liqz@caf.ac.cn (Q.L.); 2State Key Laboratory of Subtropical Silviculture, College of Forestry and Biotechnology, Zhejiang A & F University, Hangzhou 311300, China

**Keywords:** *Populus trichocarpa*, *BTB* gene family, expression patterns, hormones treatment, biotic/abiotic stresses

## Abstract

The *BTB* (Broad-complex, tramtrack, and bric-a-brac) gene family, characterized by a highly conserved BTB domain, is implicated in a spectrum of biological processes, encompassing growth and development, as well as stress responses. Characterization and functional studies of *BTB* genes in poplar are still limited, especially regarding their response to hormones and biotic/abiotic stresses. In this study, we conducted an HMMER search in conjunction with BLASTp and identified 95 *BTB* gene models in *Populus trichocarpa*. Through domain motif and phylogenetic relationship analyses, these proteins were classified into eight families, NPH3, TAZ, Ankyrin, only BTB, BACK, Armadillo, TPR, and MATH. Collinearity analysis of poplar *BTB* genes with homologs in six other species elucidated evolutionary relationships and functional conservations. RNA-seq analysis of five tissues of poplar identified *BTB* genes as playing a pivotal role during developmental processes. Comprehensive RT-qPCR analysis of 11 *BTB* genes across leaves, roots, and xylem tissues revealed their responsive expression patterns under diverse hormonal and biotic/abiotic stress conditions, with varying degrees of regulation observed in the results. This study marks the first in-depth exploration of the *BTB* gene family in poplar, providing insights into the potential roles of *BTB* genes in hormonal regulation and response to stress.

## 1. Introduction

The BTB (Broad-complex, tramtrack, and bric-a-brac) protein was initially discovered in *Drosophila*, and the BTB proteins contain a highly conserved BTB domain of approximately 115 amino acids, also known as the POZ (Pox virus and zinc) finger domain [1]. The core structure of the BTB domain comprises an α-helix hairpin structure and a β-fold structure [2]. In addition to the core domain, variations exist in both the C-terminal and N-terminal extension regions of the BTB domain, which facilitate distinct functional roles for different types of BTB proteins. DNA-binding domains within BTB proteins enable gene expression regulation, while interactions between BTB domains and other domains enhance the protein functionality [1]. BTB proteins are classified into various subfamilies based on other domains, including BTB-only, BTB–BACK, BTB–Kelch, BTB–back–Kelch, BTB–Arm, BTB–TAZ, BTB–ANK, BTB–PHR and Rho–BTB [3]. Among these subfamilies, most members of the BTB–BACK subfamily contain both the BTB domain and Kelch repeats. The Kelch repeats in the BACK proteins form a beta propeller that can interact with actin filaments [4]. The Arm, a 42-amino acid protein–protein interaction motif, was first identified in the *Drosophila* segment polarity gene, armadillo [5], and a subset of these proteins is conserved across the eukaryotic kingdoms. BTB–TAZ proteins are presented in plants. BTB–ANK is a key transcriptional co-regulator of immunity and physiology in many plant species [2,3].

BTB proteins have been extensively investigated in animals [6,7]. In recent years, the BTB proteins have been identified in many plants, including *Arabidopsis thaliana* [8], rice [9], tomato [10], *Paulownia fortunei* [11], and sugar beet [12]. BTB/POZ proteins play a very important role in various processes of plant growth and development, such as plant morphogenesis [13,14,15], secondary metabolite synthesis [16,17], plant hormone signal transduction [18,19], and stress resistance [20,21]. In Arabidopsis, the BTB protein can interact with RBX1 and CUL3 to assemble the CRL3 (E3 ubiquitin ligase complex), demonstrating the role of BTB proteins in plant growth/development and response to stress through mediating protein ubiquitination [22]. In addition, the BTB protein in conjunction with the MATH domain protein, MATH–BTB, is involved in the abscisic acid (ABA) signaling pathways and abiotic stress response [23]. In Arabidopsis, six MATH–BTB proteins (called BPM1-6) are involved in regulating seed germination as well as stomatal opening and closing [19]. BTB domain-containing proteins, ATEOL1, ATEOL2, and ATETO1, participate in the regulation of ethylene biosynthesis in plants, while the HT1 protein negatively regulates hormone-mediated inhibition of Arabidopsis germination [24]. The ATBOP1 protein containing both the BTB and ANK domains play roles in plant morphogenesis control [12]. 

During plant growth and development, plants are highly susceptible to a range of biotic/abiotic stresses, such as salinity, drought, high temperatures, insect pests, fungal infections, and so on. These dynamic environmental factors adversely influence plant normal growth and development. BTB family proteins play a pivotal role in the regulation of stress-related genes, which is integral to the plant’s adaptive response to various stressors [21]. In Arabidopsis, the BTB–TAZ protein BT2 has been observed to be responsive to a range of environmental and hormonal cues [25]. Notably, light exposure, indole-3-acetic acid (IAA), ABA, and low temperatures have been found to suppress the expression of *BT2*, whereas cytokinins (CK), methyl jasmonate (Me-JA), and hydrogen peroxide (H_2_O_2_) stimulate its expression [20]. The NPR1 protein harbors two unique protein–protein interaction motifs, an ankyrin (ANK) repeat, and a BTB/POZ domain [26]. The Sl ANK protein, characterized by its ankyrin repeats, is pivotal in responding to both biotic and abiotic stresses and is indispensable for plant growth and development in tomato [27]. In Arabidopsis, the NPR1 protein with a BTB domain exerts regulatory effects on systemic acquired resistance (SAR) and induced systemic resistance (ISR), and this dual functionality is crucial for its role in the plant stress response [28]. Overexpression of the Arabidopsis *NPR1* gene in tomatoes confers a wide-ranging resistance to various pathogens [29]. As a close relative of NPR1, the OsNPR1/NH1 protein in rice has been identified to enhance plant resistance to bacterial blight, which is caused by *Xanthomonas oryzae* pv [30]. These studies show the integral role of BTB proteins in modulating plant stress responses.

As the main worldwide afforestation tree species, especially in Northern China, poplar is the first perennial woody plant for which the whole genome sequence has been sequenced [31], and it has become a model plant for forest tree studies. Currently, the members of the *BTB* gene in poplar have not yet been reported. In this study, we conducted comprehensive bioinformatics and expression analysis of *BTB* gene in *P. trichocarpa.* A total of 95 *PtrBTB* genes were identified in the genome, and their chromosomal locations, gene pair relationships, phylogenetic comparison, and promoter *cis*-acting elements were analyzed. RT-qPCR was employed to investigate the expression pattern of the selected *BTB* genes in hormone treatments and biotic/abiotic stress conditions across different tissues. Altogether, our research details the *BTB* gene family in poplar and provides valuable information for the functional characterization of BTB proteins.

## 2. Results

### 2.1. Chromosomal Location and Homologous Gene Analysis of PtrBTB Genes

The *PtrBTB* gene family members were identified from the *P. trichocarpa* genome (v4.1) using the hidden Markov models algorithm, with the conserved BTB domain model (PF00651) as an inquiry. Using the BLASTp software (version 2.14.0) to compare with BTB proteins in Arabidopsis, a total of 95 *PtrBTB* gene models were identified in *P. trichocarpa*. For consistency, we detected the distribution of the identified *BTB* genes in chromosomes and named these genes as *PtrBTB1* to *PtrBTB95* according to their distributions and relative linear orders among their respective chromosomes. These genes were graphically aligned to the respective chromosomes, as depicted in Figure 1a. The resulting distribution map revealed an uneven distribution of 95 *PtrBTB* genes across the full spectrum of the 19 chromosomes. Among them, chromosome 11 was distinguished by having the lowest number of *PtrBTB* genes, featuring merely a single gene. Notably, chromosomes 1 and 5 each contained twelve *PtrBTB* genes, which remarkably exceeded the count on other chromosomes. 

Detailed information of each PtrBTB protein, including gene name, gene ID, chromosome location (Chr), numbers of amino acid (AA), molecular weight (MW), theoretical isoelectric point (pI), and putative subcellular localization were presented in supplementary Appendix A. These 95 PtrBTB proteins had divergent lengths, resulting in diverse isoelectric points and molecular weights. Subcellular localization analysis showed that most PtrBTB proteins were localized in both the nucleus and cytoplasm, and a few might be localized in other subcellular compartments, such as the chloroplast and mitochondria (Appendix A). No discernible pattern was observed linking the length of the chromosomes to the distribution density of *PtrBTB* genes along them. Utilizing the MCScanX software (version 2019), we successfully identified homologous genes within the *PtrBTB* gene family (Figure 1b). Our findings indicate that the *BTB* gene family in *P. trichocarpa* comprises 48 homologous gene pairs. Specifically, we observed six pairs of homologous genes between chromosomes 2 and 5 and five pairs between chromosomes 8 and 10. The non-synonymous (*Ka*)/synonymous (*Ks*) ratios of these gene pairs were both less than one, indicating that a strong purifying selection was experienced in the process of evolution (Appendix A). Among them, the *Ka* and *Ks* values for *PtrBTB75* and *PtrBTB87* of the NPH3 subfamily were the highest, while the Ankyrin subfamily’s *PtrBTB*16 and *PtrBTB*35 exhibited the maximal ratio of *Ka/Ks*.

### 2.2. Evolution Relationship of BTB Genes in Various Species

We also used the same above-mentioned method to identify the BTB family in a hybrid poplar *P. alba × P. glandulosa* and obtained 165 genes. A phylogenetic tree was constructed to explore the evolutionary relationships within the *BTB* gene family. Utilizing MEGA-X 10.2, we applied the Neighbor-Joining (NJ) method to analyze BTB proteins from *P. trichocarpa* (95 members), *P. alba × P. glandulosa* (165 members), and Arabidopsis (80 members) (Figure 2). The phylogenetic tree showed that a total of 340 BTB proteins were classified into eight subfamilies, NPH3, TAZ, Ankyrin, only BTB, BACK, Armadillo, TPR, and MATH. Among them, the NPH3 subfamily had the largest number, including 36 BTB proteins in *P. trichocarpa*, 71 in *P. alba* × *P. glandulosa*, and 32 in Arabidopsis. In contrast, the Armadillo subfamily was characterized by a lower number of BTB proteins, specifically four, each featuring eight Armadillo repeats at the C-terminus and a BTB domain at the N-terminus (Appendix A). The number of BTB proteins between poplar and Arabidopsis varied greatly in the same subfamily. The poplar BTB proteins were distributed in all eight subfamilies, and most of the poplar BTB proteins had higher homologies with Arabidopsis, suggesting a conserved functional role across these two plant species.

### 2.3. Gene Structure and Conserved Motif Analysis of PtrBTB Genes

To gain insight into the identified 95 *PtrBTB* genes, we conducted the gene structure and motif distribution analyses using the MEME suite. Up to 10 motifs were obtained and were designated motif 1 through 10 (Figure 3a). The three most conserved motifs, consisting of 41, 50, and 50 amino acids, respectively, overlapped with the BTB domain sequence (Appendix A). Conserved motif analysis revealed that various PtrBTB proteins exhibited similar types and arrangements of motif. Among them, almost all members contained motif 2, indicating that the structures of BTB are conserved. The largest subfamily NPH3 members contained eight motifs, including motifs 1, 2, 3, 4, 5, 6, 7, and 9. However, variations within the subfamilies suggest that genes among members of the same subfamily may exhibit functional diversity. 

The exon–intron pattern is a crucial feature influencing gene function diversification; therefore, the gene structure of *PtrBTB* genes was characterized accordingly (Figure 3b). The analysis revealed variability in exon numbers ranging from 1 to 19 and intron numbers ranging from 0 to 18 within the *PtrBTB* gene family. Most members of the largest subfamily NPH3 contained 4 exons and 3 introns, whereas the Armadillo subfamily, which was characterized by the highest exon–intron count, contained 19 exons and 18 introns. The genes clustered into the same clade in the phylogenetic tree had similar gene structure, suggesting a relatively recent evolutionary relationship.

### 2.4. Cis-Acting Element Prediction of PtrBTB Genes

*Cis*-acting elements are necessary clues in predicting gene functions. By attaching to the *cis*-acting component of target genes in particular organic processes, transcription elements may affect the level of gene expression [32]. The promoter regions of various *PtrBTB* genes display a variety of *cis*-acting elements, predominantly categorized into two classes. One class of these elements was implicated in the regulation of hormonal pathways, encompassing ABA, methyl jasmonate (MeJA), gibberellin (GA), and IAA, which are crucial for plant growth and development. The other class was associated with the plant response to environmental stressors, such as exposure to light, low temperatures, arid conditions, and the activation of defense mechanisms. Among them, the largest numbers of *cis*-elements were the light-response elements, and they were found in all the PtrBTB family members. Notably, the numbers of ABA response elements (151) and MeJA response elements (200) were the second highest in the promoters of *PtrBTB* genes, implying their potential involvement in plant resistance mediated by ABA and MeJA signaling pathways. Additionally, 46 of these promoters displayed a responsive nature to salicylic acid (SA), while 53 showed a GA-responsive element, and 39 were found to be responsive to auxin. Specifically, among the 63 *PtrBTB* promoters, a significant presence of MYB binding sites that were crucial for drought-inducible elements was identified (Figure 4). Collectively, these findings suggest that PtrBTB proteins may play a role in coordinating both biotic and abiotic responses in plants through hormonal signaling pathways.

### 2.5. Collinear Analysis of the PtrBTB Gene Family in Different Species

To further investigate the collinearity among the *BTB* genes, we examined the genomic synteny of *P. trichocarpa* with those of six other plant species. This group included four dicotyledons—*P. alba × P. glandulosa*, *A. thaliana*, *Solanum lycopersicum*, and *E. grandis* [33]—as well as two monocotyledons, *Oryza sativa* and *Zea mays*. As depicted in Figure 5, the number of homologous pairs between *P. trichocarpa* and these species were as follows: 87 with *P. alba × P. glandulosa*, 69 with *E. grandis*, 69 with *A. thaliana*, 68 with *S. lycopersicum*, 34 with *O. sativa*, and 26 with *Z. mays* (Figure 5). The observed phenomenon can mainly be attributed to the divergent phylogenetic relationships that distinguish dicotyledons from monocotyledons. Among them, the *PtrBTB* genes exhibited the greatest collinearity with those of *P. alba × P. glandulosa* and *E. grandis*, both of which are also dicotyledonous woody plants. Notably, *PtrBTB5*, *PtrBTB71*, *PtrBTB81*, and *PtrBTB86* were not detected in either *P. alba × P. glandulosa* or *E. grandis*, which imply that these genes might have undergone species-specific evolution. In addition, we found that a large number of *PtrBTB* genes, such as *PtrBTB19*, *PtrBTB24*, *PtrBTB31*, *PtrBTB61*, and *PtrBTB76*, had collinear relationships with three Arabidopsis genes, suggesting that these genes may play an important role in the evolution of gene families.

### 2.6. Expression Patterns of PtrBTB Genes in Different Tissues

To investigate the expression pattern of the *PtrBTB* gene family in different tissues, we used the RNA-seq to analyze the *BTB* gene expression in poplar leaves, shoots, roots, xylem, and phloem. Meanwhile, in order to ensure the accuracy of the data, we selected 11 *PtrBTB* genes for RT-qPCR validation and found that the results were consistent with the RNA-seq (Appendix A). As shown in Figure 6, the cluster analysis revealed that these genes were divided into six groups with specific expression patterns. In group Ⅰ, *PtrBTB95* was highly expressed in leaves, *PtrBTB74* was highly expressed in xylem, and *PtrBTB29* was higher in phloem, while the remaining gene had lower expression level across all tissues. In contrast, the expression level of all the *PtrBTB* genes in group Ⅱ was high, especially *PtrBTB48* and *PtrBTB75*, which had the highest expression level in leaves and shoots. A majority of the genes in group III were preferentially expressed in roots and xylem. Furthermore, *PtrBTB20*, *PtrBTB40*, and *PtrBTB83* were noted for their pronounced expression in the phloem as well. Genes in group Ⅳ had high expression levels in the roots, among which *PtrBTB90* was also highly expressed in the leaves, indicating its potential involvement in the leaf development process. Most genes in group Ⅴ exhibited high expression level in both xylem and phloem, while genes in group VI had high expression levels in the leaves and shoots, indicating these genes may function in distinct biological processes.

### 2.7. Expression Patterns of Poplar BTB Genes in Response to Exogenous Hormones

Based on RNA-seq data from previous hormone treatments and stress treatments, we identified eleven genes with significant differences, namely *PtrBTB3*, *PtrBTB4*, *PtrBTB13*, *PtrBTB16*, *PtrBTB23*, *PtrBTB38*, *PtrBTB45*, *PtrBTB47*, *PtrBTB49*, *PtrBTB80* and *PtrBTB95*, which belong to five distinct subfamilies (Appendix A). Since *P. trichocarpa* plants cannot grow well in the area of Beijing, China, we analyzed the expression of these eleven *BTB* genes in leaf tissues of *P. alba × P. glandulosa* in response to five hormone treatments (GA, ABA, SA, NAA, and JA), comparing their expression levels with those in untreated leaves of wild-type poplar. The results show that the expression levels of these *PagBTB* genes were distinctly affected by various treatments, revealing both connections and distinctions in the expression pattern among these *PagBTB* members. 

Figure 7a illustrates the expression levels of eleven *PagBTB* genes in response to JA treatment, with *PagBTB4*, *PagBTB13*, *PagBTB16*, *PagBTB38*, *PagBTB49*, and *PagBTB80* exhibiting markedly upregulated expressions. In contrast, the expression of five other *PagBTB* genes, including *PagBTB3*, *PagBTB23*, *PagBTB45*, *PagBTB47*, and *PagBTB95*, was remarkably downregulated. Upon treatment with ABA, with the exception of *PagBTB16*, *PagBTB23*, and *PagBTB49*, the expression of the other *PagBTB* genes underwent notable alterations. As depicted in Figure 7b, *PagBTB13*, *PagBTB45*, and *PagBTB80* exhibited pronounced upregulation in expression, whereas the expression levels of *PagBTB3*, *PagBTB4*, *PagBTB38*, *PagBTB47*, and *PagBTB95* were considerably downregulated. In response to GA treatment, only *PagBTB3*, *PagBTB4*, *PagBTB47*, and *PagBTB95* showed significant downregulation in expression, while the remaining seven *PagBTB* genes displayed a considerable increase in their transcript abundance (Figure 7c). The gene expression changes induced by GA were found to parallel those seen with the NAA treatment (Figure 7d), although *PagBTB3*, *PagBTB47*, and *PagBTB95* exhibited a more pronounced reduction in response to NAA. Notably, *PagBTB13* and *PagBTB23* showed obviously higher upregulation in response to GA treatment, indicating a distinct sensitivity to various plant hormones. Simultaneously, SA treatment induced the expression of *PagBTB3*, *PagBTB4*, *PagBTB13*, *PagBTB16*, *PagBTB38*, *PagBTB49*, and *PagBTB80* genes in leaves, while the expressions of *PagBTB23*, *PagBTB45*, *PagBTB47*, and *PagBTB95* genes were inhibited by SA treatment (Figure 7e).

### 2.8. Expression Patterns of Poplar BTB Genes in Response to Different Biotic/Abiotic Stress

To investigate the expression patterns of the poplar *BTB* genes under biotic and abiotic stresses, we treated leaves and roots with sodium chloride, and treated leaves with *F. solani*, while the xylem was subjected to drought stress, followed by the examination of expression changes in these above-mentioned 11 *BTB* genes in *P. alba × P. glandulosa*. In the roots treated with NaCl, the expression levels of *PagBTB3*, *PagBTB16*, *PagBTB23*, and *PagBTB47* genes were obviously suppressed, while the expression levels of the remaining genes were induced (Figure 8a). In the leaves, *PagBTB13*, *PagBTB38*, *PagBTB49*, and *PagBTB80* showed a notable increase in expression levels, whereas *PagBTB3*, *PagBTB4*, *PagBTB16*, *PagBTB23*, *PagBTB45*, *PagBTB47*, and *PagBTB95* were obviously downregulated (Figure 8b). Likewise, in response to *F. solani*-induced stress, there was significant suppression in the expression levels of *PagBTB3*, *PagBTB23*, *PagBTB45*, *PagBTB47*, and *PagBTB95*, while *PagBTB4*, *PagBTB13*, *PagBTB16*, *PagBTB38*, *PagBTB49*, and *PagBTB80* displayed a marked increase in their expression in leaves (Figure 8c). During the course of drought stress, eleven *PagBTB* genes exhibited dynamic changes in their transcriptional levels. Specifically, *PagBTB4*, *PagBTB16*, *PagBTB38*, and *PagBTB80* exhibited increased expression levels, whereas *PagBTB3*, *PagBTB23*, *PagBTB45*, *PagBTB47*, and *PagBTB49* displayed decreased expression levels. Notably, the expression of *PagBTB13* and *PagBTB95* exhibited a biphasic response, initially decreasing at D5 of the drought treatment and subsequently upregulating at D7 (Figure 8d). Overall, *PagBTB*s have different expression patterns under different types of treatments, and they may play important functions in plant resistance to hormones and biotic/abiotic stresses.

## 3. Discussion

BTB/POZ family proteins, commonly found in animals and plants, have been extensively investigated in animals. However, recent evidence suggests their essential roles in plant growth and development, encompassing diverse functions such as resistance modulation, phototropic growth regulation, ubiquitination-mediated degradation, cell cycle control, and ion channel regulation [6,34]. Nevertheless, the functional characterization of BTB/POZ proteins in poplar remains limited compared to Arabidopsis. In this study, we integrated the HMMER search method and the BLASTp method and identified 95 *BTB* genes in *P. trichocarpa* that were classified into eight subfamilies based on their conserved domains, including NPH3, TAZ, Ankyrin, only BTB, BACK, Armadillo, TPR, and MATH (Appendix A). The BTB protein family, characterized by their DNA-binding domains, is hypothesized to regulate gene expression. Moreover, the interaction of BTB domain with additional domains expands the functional range of the BTB proteins [1]. Among them, NPH3 is the largest subfamily and contains the most BTB members in poplar. Recently, the NPH3, exclusive to higher plants, has been found to engage with the blue light receptor phot1, acting as a key substrate adapter for the CULLIN3-dependent E3 ubiquitin ligase [35]. It is evident that the NPH3 subfamily plays a pivotal role in the blue light-mediated phototropic responses, serving as an essential element in this biological process. We also found that the high degree of conservation of BTB family members allows genes localized in the same subfamily to have the same or similar functions. Gene structure and motif analysis further confirmed a close relationship among the poplar *BTB* genes within the same subgroup, indicating consistent exons counts and motif types that align with observed distribution patterns in the phylogenetic tree. This intriguing aspect warrants further investigation to elucidate its underlying mechanisms and potential applications. 

The expression of genes is mostly related to the *cis*-elements in the promoter region [36]. In Arabidopsis, the *cis*-acting element DRE (Dehydration-responsive element) plays a crucial role in regulating gene expression under conditions of dehydration and low temperature [37]. Analysis of *cis*-elements in the promoter region revealed a significant presence of plant hormone-related and stress-responsive elements within the promoters of the *BTB* genes in *P. trichocarpa*, indicating their potential involvement in related functional expression. Notably, this investigation has uncovered a range of *cis*-elements within *PtrBTB* promoters involved in responses to light exposure, hormonal stimuli, and environmental stress (Figure 4). It is worth noting that light-responsive elements are ubiquitous and most abundant, aligning with the findings from various species including grape [38], sugar beet [12], and tomato [10]. This result suggests that the transcriptional control of PtrBTB is likely influenced by light signals across a broad spectrum of plant species. In addition, four distinct poplar *BTB* genes, namely *PtrBTB4*, *PtrBTB44*, *PtrBTB54*, and *PtrBTB82*, have been identified to contain *cis*-elements associated with wound response. This discovery is novel as these elements have not been previously reported in species such as soybeans, apples, and grapes, suggesting a potential role for these genes in the self-healing process and enhanced vitality of poplar trees.

The cross-species collinearity analysis indicated a closer evolutionary relationship between *P. trichocarpa* and dicotyledons compared to monocotyledons (Figure 5). This observation hints at a potential divergence in the evolution of *BTB* genes in conjunction with the evolutionary split between dicotyledons and monocotyledons. Collectively, the bioinformatic assessments indicate congruence in phylogenetic and structural characteristics of the *PtrBTB* gene family across various species, suggesting analogous functional roles of *BTB* genes and underscoring their conserved biological significance. In addition, previous reports have shown that the BTB protein can promote plant resistance to various abiotic and biological stresses. The expression of the *AtBT4* gene in Arabidopsis is upregulated, thereby enhancing the plant resistance to *Botrytis griseus* through the ethylene (ET)/JA pathway [39]. Additionally, under drought conditions, the expression of the *MdBT2* gene in apple is downregulated, resulting in the activation of *NAM*, *ATAF1/2*, *CUC1/2* (*NAC*). These genes, originally inhibited by MdBT2 protein, subsequently pass through acid signaling pathways that regulate reactive oxygen species (ROS) levels and participate in plant stress response [40]. Furthermore, following inoculation with potato virus Y (PVY), tobacco initiates a defense response involving active hormones such as IAA, SA, and JA, along with ROS, suggesting the potential regulatory roles of the BTB/POZ family members in the response of tobacco to PVY [41]. Therefore, we subjected the plant material to drought, salt, *F. solani* stress, and various hormones treatments. Based on prior RNA-seq data, we chose eleven *BTB* genes that exhibited significant expression changes under various treatment conditions. Specifically, the genes *PtrBTB45*, *PtrBTB47*, and *PtrBTB95* belong to the NPH3–BTB subfamily, *PtrBTB13* and *PtrBTB38* are members of the BACK–BTB subfamily, and *PtrBTB49* and *PtrBTB80* are associated with the TAZ–BTB subfamily. *PtrBTB16* is a part of the Ank–BTB subfamily, whereas *PtrBTB3*, *PtrBTB4*, and *PtrBTB23* are classified under the only-BTB subfamily, which lacks additional domains. 

The RT-qPCR analysis across the three treatments revealed that a majority of these *PagBTB* genes demonstrated a significant response to the applied treatments. Following hormonal stimuli and environmental perturbations, the genes exhibited diverse temporal expression patterns, and the individual genes showed tissue-specific modulation, emphasizing the complexity of gene expression in response to stimuli. Upon hormonal intervention, *PagBTB13* and *PagBTB80* were characterized by a significant upregulation in their expression levels, in contrast with the diminished expression levels of *PagBTB47* and *PagBTB95*, illustrating a divergent gene expression response to hormonal stimuli (Figure 7). Furthermore, following NaCl treatment, *PagBTB4*, *PagBTB45*, and *PagBTB95* showed a significant upregulation in the root tissues but notable downregulation in the leaf tissues (Figure 8). These observations suggest that poplar *BTB* genes are responsive to stress stimuli and exhibit tissue-specific expression patterns, with distinct responses in roots compared to leaves. Collectively, our findings from promoter analysis and gene expression studies under stress conditions suggest a pivotal role for the BTB proteins in plant response to both biotic and abiotic stressors [42].

## 4. Materials and Methods

### 4.1. Plant Materials and Treatment

The experimental materials used in this study were from the State Key Laboratory of Tree Genetics and Breeding, Chinese Academy of Forestry, Beijing, China. Sterile *Populus alba* × *Populus glandulosa* (84K) plants were grown on one-half Murashige and Skoog (Sigma-Aldrich, Darmstadt, Germany) media containing 3% sucrose and 0.6% agar and maintained at 23 °C under a 16 h light/8 h dark photoperiod. The plantlets were grown on humus soil with a 16/8 h day/night cycle at 25 °C in the greenhouse. For the gene expression profiling analysis, the *P. alba* × *P. glandulosa* plants were simultaneously subjected to various stresses or hormone treatments. The culture seedlings of 1-month-old were transplanted into 1/2 MS liquid medium containing 3% (*w*/*v*) sucrose and pre-cultured for 16 h. Samples were then treated with 1/2 MS liquid medium containing 100 μM 1-Naphthaleneacetic acid (NAA), 100 μM ABA, 100 μM gibberellic acid (GA), 100 μM SA, 100 μM GA, 200 mM NaCl, and 10^7^ spores/mL *Fusarium solani* (*Mart.*) *Sacc* (*F. solani*), respectively.

Following these treatments, the leaves and roots were collected as experimental materials under the NaCl treatment; the leaves were collected as experimental materials in the other treatments. Then, the tissues were harvested after various treatment time points (0, 2, 4, 6, 12, 24 h), immediately frozen in liquid nitrogen and then stored at −80 °C for RNA preparation. Pilot studies on drought stress have indicated that applying both 5-day (D5) and 7-day (D7) drought treatments could potentially induce the most significant and extensive genetic alterations [43]. Consequently, we cultivated 3-month-old poplar 84K in a controlled environment to a stage of maturity suitable for experimentation and subjected them to the aforementioned drought conditions. Subsequently, samples of differentiating xylem from the stems of both well-irrigated (control group, devoid of drought stress) and the drought-affected plants were harvested. These samples were immediately frozen in liquid nitrogen and stored at −80 °C for RNA extraction. Each sample was collected in triplicate as biological replicates. 

### 4.2. Identification and Annotation of BTB Proteins in P. trichocarpa

Genome data and genome annotation files of *P. trichocarpa* (v4.1), Arabidopsis (TAIR10) were extracted from the Phytozome database (https://phytozome-next.jgi.doe.gov/ (accessed on 24 December 2023)), and the files of *P. alba × P. glandulosa* (v3.1) were downloaded from the website (https://doi.org/10.6084/m9.figshare.12369209 (accessed on 24 December 2023)) [44]. The Arabidopsis BTB protein sequences were utilized as queries for BLASTp searches against the *P. trichocarpa* and the *P. alba × P. glandulosa genomes*. The following parameters were used: −evalue 10^−5^, −best hit overhang 0.25, and −max target seqs 5, with the searches conducted in the NCBI database (https://www.ncbi.nlm.nih.gov/ (accessed on 24 December 2023)). Furthermore, the HMM file (PF00651) of the conserved domain was downloaded from the Pfam database (http://pfam-legacy.xfam.org/ (accessed on 24 December 2023)). A Hidden Markov Model profile of the HMMER 3.0 software was employed to search for putative BTB in *P. trichocarpa* and *P. alba × P. glandulosa*, with an E-value threshold of 10^−5^. The results obtained by the two methods were integrated with those of previous studies to avoid repetition. After the protein sequences were aligned and trimmed by ClustalW, the protein sequences of 95 BTB proteins in *P. trichocarpa*, 165 BTB proteins in poplar 84K, and 80 BTB proteins in Arabidopsis were used to construct the phylogenetic tree by the MEGA-X 10.2 Neighbor-Joining (NJ) method, and the parameters were p-distance model and 1000 bootstrap replicates. Subsequently, a phylogeny of the BTB proteins was generated on the iTOL web browser (https://itol.embl.de/ (accessed on 30 December 2023)). Furthermore, the physical and chemical parameters of PtrBTB proteins were determined using the ExPASy website (http://web.expasy.org/protparam/ (accessed on 30 December 2023)). Subcellular locations of PtrBTB proteins were predicted using the WoLF PSORT ProtParam online analysis tool (https://www.genscript.com/wolf-psort.html (accessed on 30 December 2023)).

### 4.3. Chromosomal Location and Collinearity Analysis

GTF annotations of the *P. trichocarpa* genome were mapped to each *PtrBTB* gene and to its corresponding chromosomal location using the Gene Location Visualize module of TBtools software (version 2.019) [45]. The genome sequence files of *P. trichocarpa*, *P. alba × P. glandulosa*, *Eucalyptus grandis*, *A. thaliana*, *Oryza sativa*, *Solanum lycopersicum*, and *Zea mays* were aligned and analyzed using the MCScanX software (version 2019). Gene collinearity analysis was used to determine their conservative relationships and visualized by TBtools (version 2.019). We also used the KaKs_Calculator software (version 3.0) to calculate the ratio of non-synonymous substitution and synonymous substitution (*Ka/Ks*) for duplication gene pairs with the NG method [46].

### 4.4. Analysis of Domains, Gene Structures, Motifs and Cis-Acting Elements

Conserved domains and motifs of BTB proteins were identified using the SMART (http://smart.embl-heidelberg.de/ (accessed on 14 March 2024)), GSDS (https://gsds.gao-lab.org/Gsds_help.php (accessed on 18 March 2024)), and MEME (http://meme-suite.org/tools/meme (accessed on 25 March 2024)) websites. The gff3 file from the *P. trichocarpa* whole genome database was utilized to identify and annotate gene structures, including the intron, exon, and UTR regions. The 2000 bp upstream regions of all the *PtrBTB* genes were extracted for analysis of promoter *cis*-acting elements through the PlantCARE online database (http://bioinformatics.psb.ugent.be/webtools/plantcare/html/ (accessed on 26 March 2024)) [47]. All those generated files were visualized using the TBtools software (version 2.019) [45].

### 4.5. Expression Pattern Analysis of PtrBTB Genes

The RNA-seq data under different treatments in *P. trichocarpa* were obtained from previous studies [43,48,49]. The corresponding expression data were downloaded from the National Center for Biotechnology Information Gene Expression Omnibus database (https://www.ncbi.nlm.nih.gov/geo/ (accessed on 18 March 2024)) (accession number: GSE81048, GSE81077, GSE109609) and the BIGD Genome Sequence Archive (https://bigd.big.ac.cn (accessed on 18 March 2024)) under accession number CRA006695. Transcript levels were normalized based on fragments per kilobase of transcript per million fragments (FPKM) with Cufflinks (v. 2.1.1) with default options. Heatmaps were plotted using TBtools (version 2.019) [45].

### 4.6. RNA Isolation and Quantitative Real-Time PCR

Total RNAs were isolated using the RNeasy Plant Mini Kit (QIAGEN, Redwood, CA, USA), and mRNAs were reverse-transcribed into cDNA using the Prime Script TM RT Reagent Kit with a gDNA Eraser (TaKaRa, Dalian, China). RT-qPCR was conducted using the CFX96 Real-Time PCR (Bio-Rad, Hercules, CA, USA) with SYBR Green Premix ExTagII (TaKaRa, Dalian, China). The PCR program was as follows: initial incubation at 94 °C for 5 min; 40 cycles of 94 °C for 10 s, 58 °C for 15 s, and 72 °C for 30 s. Data were gathered during the extension step. Melting-curve acquisition and analyses were performed on the cycler, with relative expression changes of genes analyzed using the 2^−ΔΔCT^ method. *Actin* was used as the internal reference. The gene-specific primers are listed in Appendix A. Reactions were performed with three biological replicates, each with three technical replicates.

## 5. Conclusions

In our research, we have successfully identified 95 *PtrBTB* genes in *P. trichocarpa* genome and classified them as eight subfamilies. The chromosomal mapping of these genes exposed an uneven distribution, with 95 *PtrBTB* genes scattered across nineteen chromosomes. Structural and motif analyses revealed remarkable consistency in gene architecture and motif composition across all members within each cluster. Furthermore, collinearity analysis uncovered a greater prevalence of collinear genes between the woody plant poplar and *E. grandis*, suggesting a potential gene divergence coinciding with the evolution of herbaceous and woody species. A detailed examination of the gene promoters revealed a significant presence of the *cis*-elements associated with hormones and biotic/abiotic stress responses, indicating a probable involvement of these genes in the stress adaptation mechanisms. RT-qPCR analysis further substantiated the responsiveness of the eleven *PagBTB* genes to a spectrum of hormonal and stress stimuli. This comprehensive analysis enhances our understanding of the *BTB* gene family in poplar and paves the way for future studies aimed at enhancing stress resilience in plants.

## Figures and Tables

**Figure 1 ijms-25-09048-f001:**
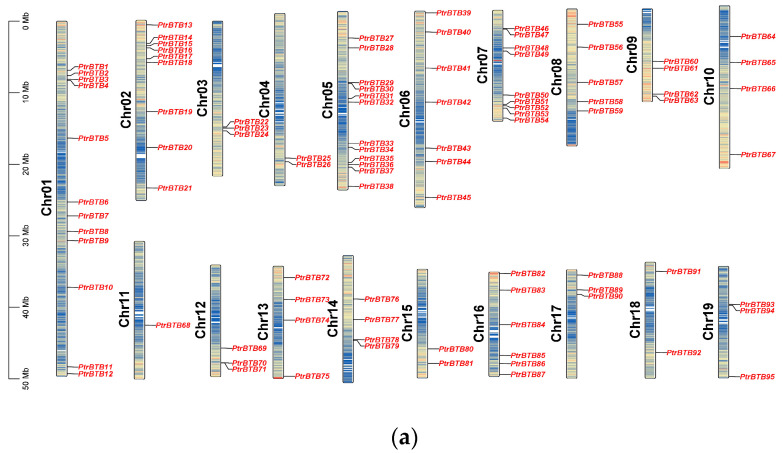
Chromosome distribution and synteny analysis of *BTB* genes in *P. trichocarpa*. (**a**) The chromosomal mapping of *PtrBTB* genes across all 19 chromosomes of *P. trichocarpa*. (**b**) Comparative analysis of the distribution and syntenic relationships within the *PtrBTB* gene family, with syntenic gene pairs visually represented by blue connecting lines.

**Figure 2 ijms-25-09048-f002:**
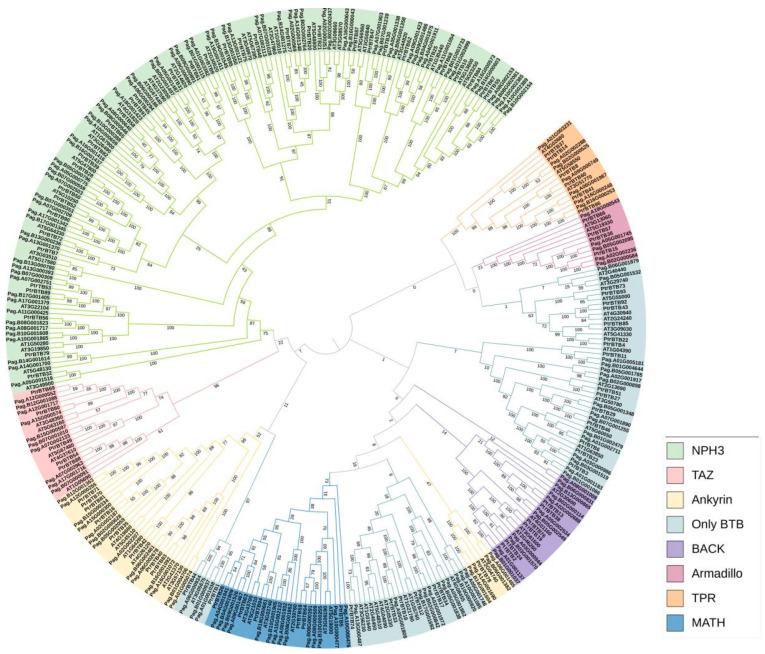
A phylogenetic tree of BTB proteins from three plant species, *P. trichocarpa*, *P. alba* × *P. glandulosa*, and Arabidopsis. The analysis shows that 340 BTB proteins are classified into eight subgroups: MATH, Armadillo, BTB-only, TAZ, BACK, Ankyrin, TPR, and NPH3. The phylogenetic tree was constructed using MEGA-X 10.2 software using the neighbor-joining method at 1000 bootstrap replicates.

**Figure 3 ijms-25-09048-f003:**
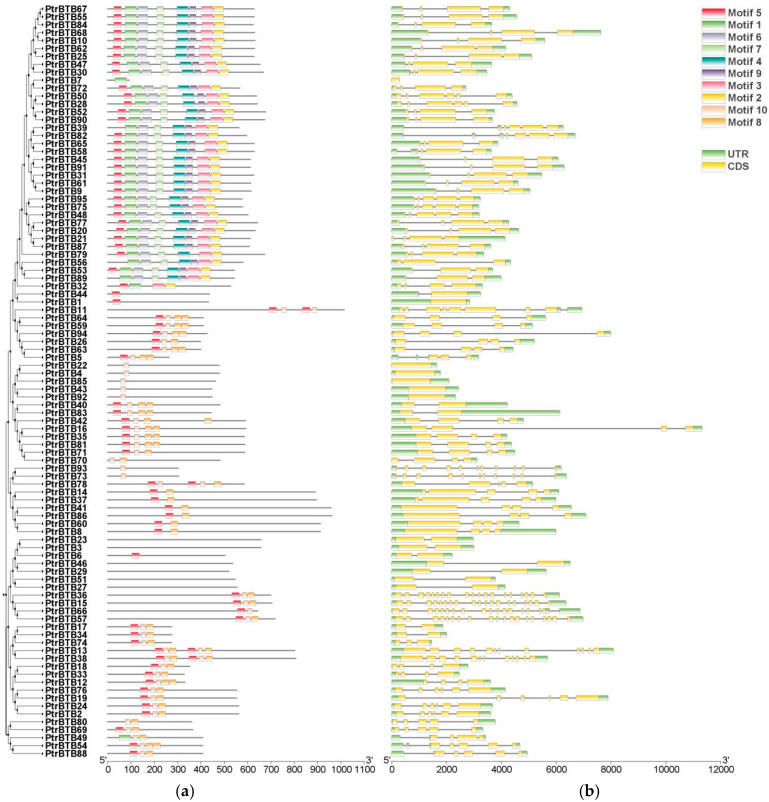
Analysis of the conserved motifs and gene structures within the *PtrBTB* gene family. The clustering is performed according to the results of phylogenetic analysis. (**a**) Conserved motifs within *PtrBTB* proteins, identified by the MEME tool (version 5.5.6), yielding 10 distinct motifs, labeled as Motif 1 through 10. 100 aa is indicated by the scale bar. (**b**) Gene structure analysis of *PtrBTB* genes revealed the organization of UTR, intron, and exon regions, with UTR in green, exons in yellow, and introns in grey. The scale bar corresponds to 2 kb.

**Figure 4 ijms-25-09048-f004:**
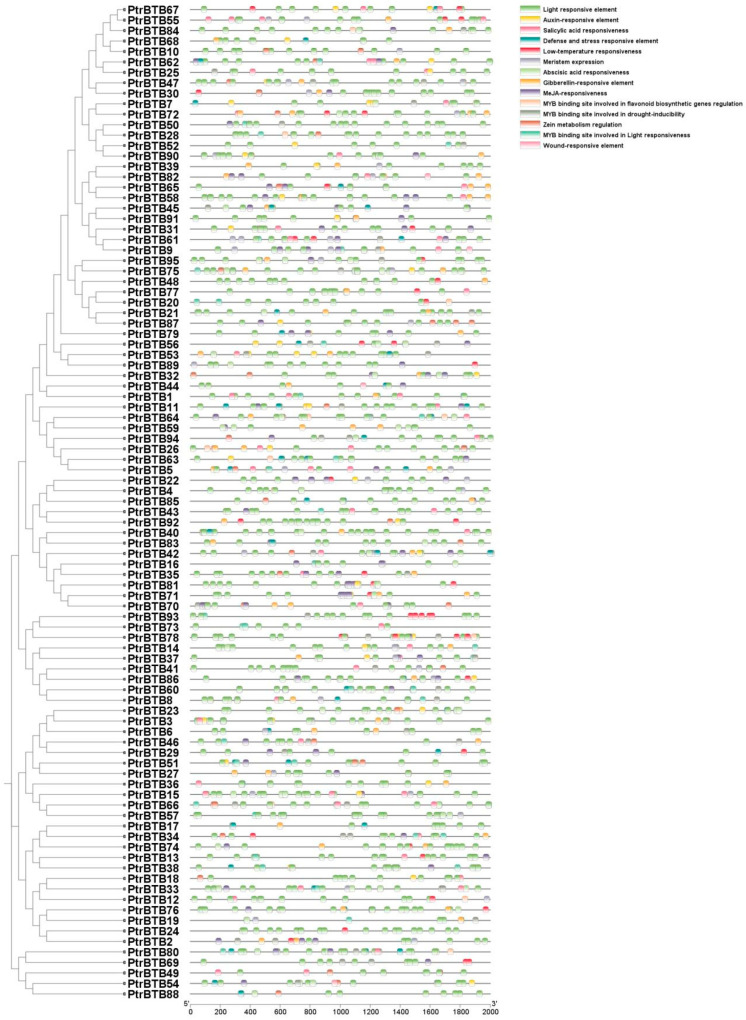
Analysis of *cis*-elements in the 2000 bp upstream promoter of *PtrBTB* genes. The clustering is performed according to the results of phylogenetic analysis. Within the predicted promoter region of the *PtrBTB* genes, a total of 14 distinct regulatory motifs have been identified. These motifs are distinguished by a spectrum of colors, each representing a unique class of transcription factor binding sites.

**Figure 5 ijms-25-09048-f005:**
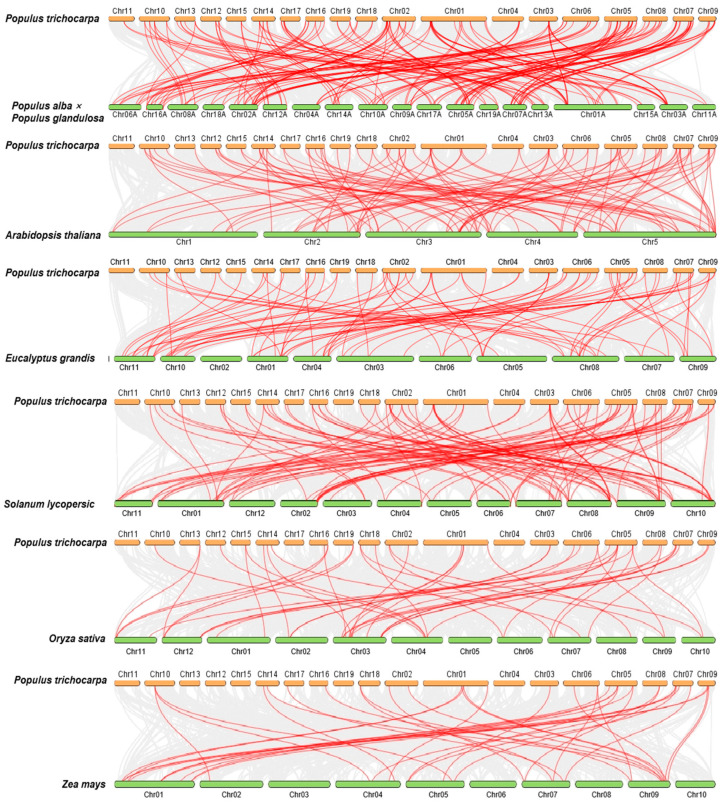
Collinear analysis of *PtrBTB* genes from *P. trichocarpa* with four dicotyledons (*P. alba × P. glandulosa*, *A. thaliana*, *S. lycopersicum*, and *E. grandis*) and two monocotyledons (*O. sativa* and *Z. mays*).

**Figure 6 ijms-25-09048-f006:**
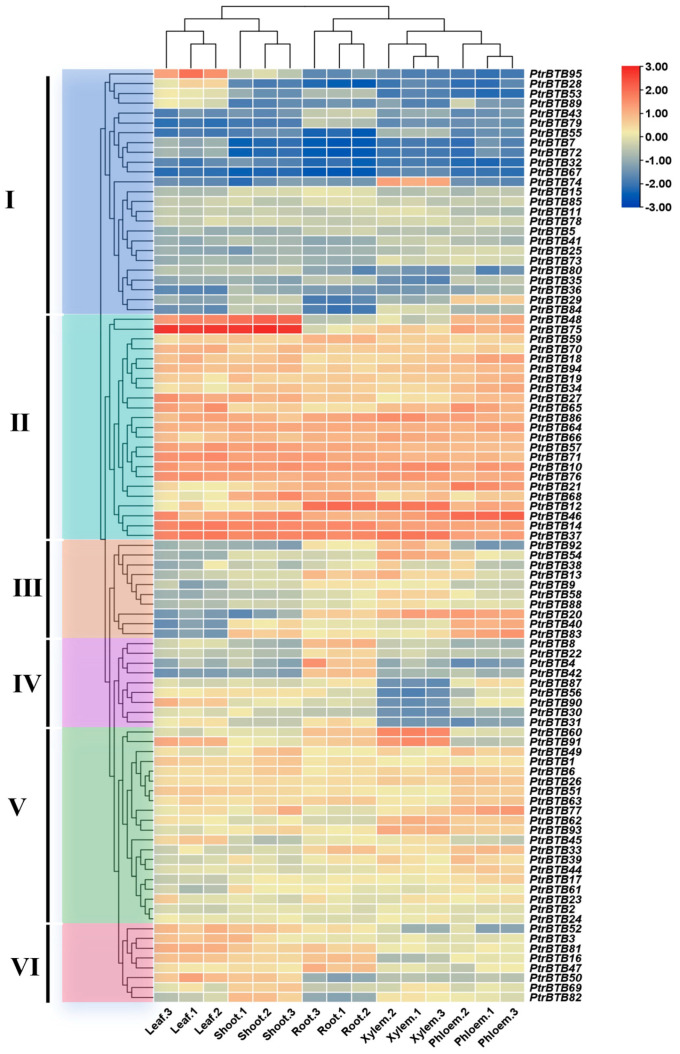
Expression profiles of *PtrBTB* genes in different tissues, shoots, roots, leaves, xylem and phloem. Ⅰ−VI represent different clusters. All data of RNA−seq analysis was deposited in GEO (accession number: GSE81077). The color scale represents the FPKM values normalized by log_2_FPKM. Red represents highly expressed genes and blue represents low expressed genes.

**Figure 7 ijms-25-09048-f007:**
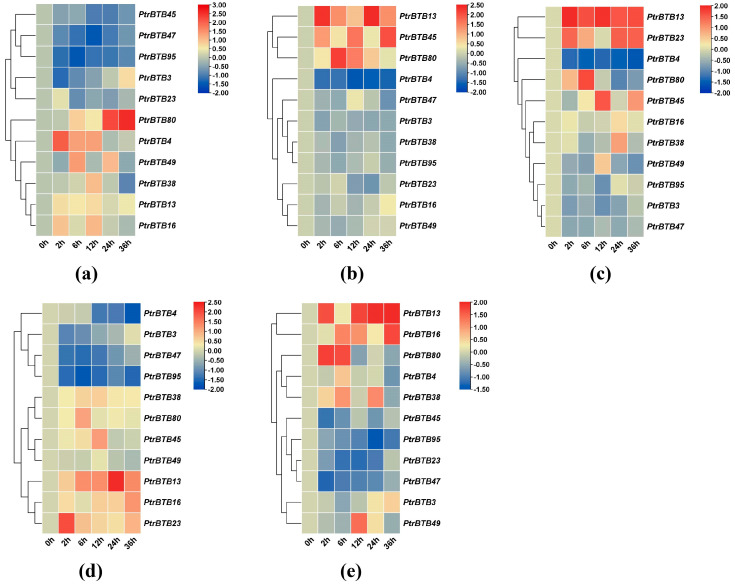
Expression profiles of *BTB* genes under exogenous hormones in *P. alba × P. glandulosa*. Color scale represents log_2_ expression values, red represents highly expressed genes and blue represents low expressed genes. (**a**) JA treatment, (**b**) ABA treatment, (**c**) GA treatment, (**d**) NAA treatment, (**e**) SA treatment. All samples were collected from leaf tissues at specified time points, with three biological replicates for each treatment. Error bars indicate ± SE of the means (*n* = 3).

**Figure 8 ijms-25-09048-f008:**
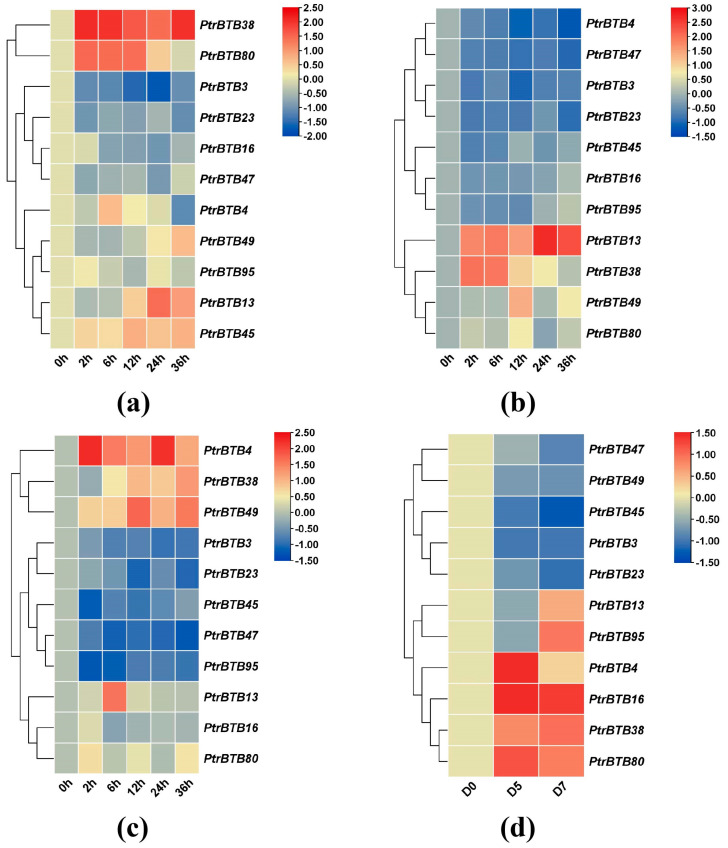
Expression profiles of *PagBTB* genes under different biotic/abiotic stress in poplar 84K. Color scale represents log_2_ expression values, red represents highly expressed genes and blue represents low expressed genes. (**a**) NaCl−root, (**b**) NaCl−leaf, (**c**) *F. solani* stress−leaf, (**d**) Drought treatment−xylem. All samples were collected at the indicated time intervals from three biological replicates of each treatment. Error bars indicate ± SE of the means (*n* = 3).

## Data Availability

All data are contained within the article and Appendix A.

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
