# Peer review of "Genome-Wide Characterization of the BTB Gene Family in Poplar and Expression Analysis in Response to Hormones and Biotic/Abiotic Stresses"

_ijms, 2024, doi:10.3390/ijms25169048_

Round 1

Reviewer 1 Report

Comments and Suggestions for Authors

Dear Authors,

Reviewer comments ijms-3147343

The manuscript entitled „Genome-wide characterization of the BTB gene family in poplar and expression analysis in response to hormones and biotic/abiotic stresses“ represents a comprehensive study on BTB protein family in poplar (Populus trichocarpa) including their structure, domain motifs (8 subfamilies of BTB proteins were distinguished according to their domain motifs), chromosomal distribution and synteny analysis, a phylogenetic analysis, and an expression analysis of selected PtBTB transcripts under selected stress treatments. The manuscript thus represents a complex and comprehensive overview on BTB protein family in poplar. I can recommend the manuscript for publication in IJMS; however, I have some important comments on the present manuscript.

1/ In section 4.1. Plant materials and treatment, the source of experimental Populus alba × populus glandulosa (84K) plants used for the analyses has to be given, i.e., from which institution the plants were obtained.

2/ In Results, Figure 2 providing a phylogenetic analysis of BTB proteins, the algorithm used for the phylogenetic tree construction (neighbour-joining method) and explanation of the numbers at nodes (values per 1,000 replicates) has to be added to the figure legend.

3/ Terminology: Correct the typing error in the protein domain name „Armadillo“ (not „Armdaillo“).

4/ Formal comments on the text related to English language and style:

Introduction, line 48: Use present perfect tense instead of the present tense in the statement: „In recent years, the BTB proteins have been identified in many plants including Arabidopsis thaliana…..“

Results, line 99: Correct the typing error in the plant name „Arabidopsis“ (not „Arabidopiss“).

Results, line 106: Correct the word form „Among them“ (not „Amone them“).

Results, line 106: replace the word „the least“ with „the lowest“ in the statement: „Among them, chromosome 11 is distinguished by having the lowest number of PtrBTB genes, featuring merely a single gene.“

Results, line 166: Correct the typing error in the term „Armadillo subfamily“ (not „Armdaillo“).

Figure 3 legend, line 175: Correct the typing error in the term „Gene structure analysis“ (not „Genetructure analysis“).

Results, line 189: Correct the typing error in the word „family“ (not „fmaily“).

Results, line 216: Modify the word „it“ to „its“ in the statement: „….indicating its potential involvement in leaf.“

Results, line 219: Modify the verb form „may functioning“ to „may function“ in the statement: „….indicating these genes may function in distinct biological processes.“

Discussion, line 337: Correct the typing error in the word „poplar“ (not „popolar“).

Discussion, line 352: Correct the statement: „plant resistance to Botrytis griseus“ (not „plant resistan to Botris griseus“).

Discussion, line 366: Add „a“ preceding the word „part“ in the statement: „PtrBTB16 is a part of the Ank-BTB subfamily…..“

Discussion, line 379: Correct the typing error in the plant name „poplar“ )not „popolar“).

Final recommendation: Accept after a minor revision.

Comments on the Quality of English Language

Dear Authors,

Reviewer comments ijms-3147343

The manuscript entitled „Genome-wide characterization of the BTB gene family in poplar and expression analysis in response to hormones and biotic/abiotic stresses“ represents a comprehensive study on BTB protein family in poplar (Populus trichocarpa) including their structure, domain motifs (8 subfamilies of BTB proteins were distinguished according to their domain motifs), chromosomal distribution and synteny analysis, a phylogenetic analysis, and an expression analysis of selected PtBTB transcripts under selected stress treatments. The manuscript thus represents a complex and comprehensive overview on BTB protein family in poplar. I can recommend the manuscript for publication in IJMS; however, I have some important comments on the present manuscript.

1/ In section 4.1. Plant materials and treatment, the source of experimental Populus alba × populus glandulosa (84K) plants used for the analyses has to be given, i.e., from which institution the plants were obtained.

2/ In Results, Figure 2 providing a phylogenetic analysis of BTB proteins, the algorithm used for the phylogenetic tree construction (neighbour-joining method) and explanation of the numbers at nodes (values per 1,000 replicates) has to be added to the figure legend.

3/ Terminology: Correct the typing error in the protein domain name „Armadillo“ (not „Armdaillo“).

4/ Formal comments on the text related to English language and style:

Introduction, line 48: Use present perfect tense instead of the present tense in the statement: „In recent years, the BTB proteins have been identified in many plants including Arabidopsis thaliana…..“

Results, line 99: Correct the typing error in the plant name „Arabidopsis“ (not „Arabidopiss“).

Results, line 106: Correct the word form „Among them“ (not „Amone them“).

Results, line 106: replace the word „the least“ with „the lowest“ in the statement: „Among them, chromosome 11 is distinguished by having the lowest number of PtrBTB genes, featuring merely a single gene.“

Results, line 166: Correct the typing error in the term „Armadillo subfamily“ (not „Armdaillo“).

Figure 3 legend, line 175: Correct the typing error in the term „Gene structure analysis“ (not „Genetructure analysis“).

Results, line 189: Correct the typing error in the word „family“ (not „fmaily“).

Results, line 216: Modify the word „it“ to „its“ in the statement: „….indicating its potential involvement in leaf.“

Results, line 219: Modify the verb form „may functioning“ to „may function“ in the statement: „….indicating these genes may function in distinct biological processes.“

Discussion, line 337: Correct the typing error in the word „poplar“ (not „popolar“).

Discussion, line 352: Correct the statement: „plant resistance to Botrytis griseus“ (not „plant resistan to Botris griseus“).

Discussion, line 366: Add „a“ preceding the word „part“ in the statement: „PtrBTB16 is a part of the Ank-BTB subfamily…..“

Discussion, line 379: Correct the typing error in the plant name „poplar“ )not „popolar“).

Final recommendation: Accept after a minor revision.

Author Response

Reviewer1

Comments and Suggestions for Authors

The manuscript entitled “Genome-wide characterization of the BTB gene family in poplar and expression analysis in response to hormones and biotic/abiotic stresses” represents a comprehensive study on BTB protein family in poplar (Populus trichocarpa) including their structure, domain motifs (8 subfamilies of BTB proteins were distinguished according to their domain motifs), chromosomal distribution and synteny analysis, a phylogenetic analysis, and an expression analysis of selected PtBTB transcripts under selected stress treatments. The manuscript thus represents a complex and comprehensive overview on BTB protein family in poplar. I can recommend the manuscript for publication in IJMS; however, I have some important comments on the present manuscript.

Response: We thank the reviewer for the comment.

Comments 1: In section 4.1. Plant materials and treatment, the source of experimental Populus alba × populus glandulosa (84K) plants used for the analyses has to be given, i.e., from which institution the plants were obtained.

Response1: Thank you for pointing this out. We agree with this comment. All experimental materials are from the State Key Laboratory and have been accurately described in line 417.

Comments 2: In Results, Figure 2 providing a phylogenetic analysis of BTB proteins, the algorithm used for the phylogenetic tree construction (neighbour-joining method) and explanation of the numbers at nodes (values per 1,000 replicates) has to be added to the figure legend.

Response2: Thank you for pointing this out. We agree with this comment. The relevant information has been added in the legend in Figure 2. 

Comments 3: Terminology: Correct the typing error in the protein domain name “Armadillo” (not “Armdaillo”).

Response3: Thank you for pointing this out. We have checked the entire manuscript and corrected a typographical errors in line 18, 179, and 341.

Comments 4: Formal comments on the text related to English language and style:

Response4: All of them have been carefully modified according to the reviewer's opinions.

Introduction, line 48: Use present perfect tense instead of the present tense in the statement: “In recent years, the BTB proteins have been identified in many plants including Arabidopsis thaliana…..”

Response: We have checked the entire manuscript and corrected a grammatical errors in line 49.

Results, line 99: Correct the typing error in the plant name “Arabidopsis” (not “Arabidopiss”).

Response: We have checked the entire manuscript and corrected a typographical error in line 104.

Results, line 106: Correct the word form “Among them” (not “Amone them”).

Response: We have checked the entire manuscript and corrected a typographical error in line 110.

Results, line 106: replace the word “the least” with “the lowest” in the statement: “Among them, chromosome 11 is distinguished by having the lowest number of PtrBTB genes, featuring merely a single gene.”

Response: We have checked the entire manuscript and corrected a typographical error in line 110.

Results, line 166: Correct the typing error in the term “Armadillo subfamily” (not “Armdaillo”).

Response: We have checked the entire manuscript and corrected a typographical error in line 179.

Figure 3 legend, line 175: Correct the typing error in the term “Gene structure analysis” (not “Genetructure analysis”).

Response: We have checked the entire manuscript and corrected a typographical error in line 189.

Results, line 189: Correct the typing error in the word “family” (not “fmaily”).

Response: We have checked the entire manuscript and corrected a typographical error in line 203.

Results, line 216: Modify the word “it” to “its” in the statement: “….indicating its potential involvement in leaf.”

Response: We have checked the entire manuscript and corrected a typographical error in line 256.

Results, line 219: Modify the verb form “may functioning” to “may function” in the statement: “….indicating these genes may function in distinct biological processes.”

Response: We have checked the entire manuscript and corrected a grammatical errors in line 259.

Discussion, line 337: Correct the typing error in the word “poplar” (not “popolar”).

Response: We have checked the entire manuscript and corrected a typographical error in line 368.

Discussion, line 352: Correct the statement: “plant resistance to Botrytis griseus” (not “plant resistan to Botris griseus”).

Response: We have checked the entire manuscript and corrected a grammatical errors in line 383.

Discussion, line 366: Add “a” preceding the word “part” in the statement: “PtrBTB16 is a part of the Ank-BTB subfamily…..”

Response: We have checked the entire manuscript and corrected a grammatical errors in line 398.

Discussion, line 379: Correct the typing error in the plant name “poplar” )not “popolar”).

Response: We have checked the entire manuscript and corrected a grammatical errors in line 410.

Reviewer 2 Report

Comments and Suggestions for Authors

Comments to Authors

In this study entitled “Genome-Wide Characterization of the BTB Gene Family in Poplar and Expression Analysis in Response to Hormones and Biotic/Abiotic Stresses”, the authors surveyed 95 PtrBTB genes, and analyzed their characterizations, phylogenetic analysis, synteny analysis and expression patterns during normal condition, as well as the responses to hormones and biotic/abiotic stresses. This manuscript is an original work, and their job provides valuable reference for studying the function of PtrBTB genes. In a word, this manuscript be suitable for publication in international journal of molecular sciences.

Here, giving a few comments or suggestions as follows:

1.     Add the Ka/Ks analysis to PtrBTB genes. Which gene has the largest Ka/Ks, Ks and Ka? How about the Ka/Ks, Ks and Ka in subfamilies? is there a correlation with gene expression level?

2.     Please ensure that gene and species names were written in italics and protein names were written in upright letters, checked them carefully throughout the paper.

3.     In fact, promoter cis-elements analysis could be added to correlate with gene expressions patterns under abiotic stress and phytohormone treatments.

4.     Some data used by the authors were all from public data, and there is a lack of verification to this data. For example, the authors should do a simple qRT-PCR analysis for some genes to increase the value of expression patterns.

5.     Why did the authors not name the gene names of PtrBTB genes according to their homologous gene in A. thaliana? Using their homologous gene in A. thaliana to name genes, the readers can more clearly understand the gene correspondence, as well as the variations of gene copy number and others.

6.     The expression level was Log2(FPKM values), but not Log2(FPKM+1), is it true?

7.     Line 462, Populus trichocarpa should be P. trichocarpa.

8.     “significantly” need a statistical test, otherwise, use “validly”, “importantly”, “availably”, “remarkably” or “obviously”. Please checked them carefully.

9.     Detail descriptions should be added for NJ tree, for example, the parameters for phylogenetic tree, p-disctance?

10.  What did the method to construct the phylogenetic tree in figure 3 and figure 4? And what sequences did the authors use? Please explain.

11.  In figure 5, the FPKM values ranged from 2 to -2, can gene expression be negative? Please check.

12. Switch Parts 2.5 and 2.6.

13.  Improve the quality of figure 7 and figure 8.

14.  Add the methods to BLASTp in Materials and Methods?

15. Please check all the references to ensure their consistency.

Round 2

Reviewer 2 Report

Comments and Suggestions for Authors

I am very glad that the authors adopted my suggestion and made relevant modifications. The current version is satisfactory, but some minor problems still need attentions. List some as following:

1.      Ka, Ks, Ks/Ks should be italic

2.      Provide the parameters to identified candidate genes using BLASTP

3.      Provide the model to calculate Ks, ka

Author Response

Comments 1: Ka, Ks, Ka/Ks should be italic.

Response1: Thank you for pointing this out. We agree with this comment. Therefore, Ka, Ks, Ka/Ks have been italic, in line 126, 128, 130 and 472 of the paper.

Comments 2: Provide the parameters to identified candidate genes using BLASTp.

Response2: Thank you for pointing this out. The parameters of BLASTp have been added to the material method in line 445 of the paper. The parameters used were: -evalue 1e-5, -best hit overhang 0.25, and -max target seqs 5, with the searches conducted in the NCBI database (https://www.ncbi.nlm.nih.gov/).

Comments 3: Provide the model to calculate Ks, Ka.

Response3: Thank you for pointing this out. The method for calculating Ks, Ka were NG method, which has been added to the material method in line 470 of the paper.
